# Supernatant of platelet-*Klebsiella pneumoniae* coculture induces apoptosis-like death in *Klebsiella pneumoniae*

Wenting Wang,[1,2] Yaozhen Chen,[1] Yutong Chen,[1] Erxiong Liu,[1] Jing Li,[2] Ning An,[1] Jinmei Xu,[1] Shunli Gu,[1] Xuan Dang,[1] Jing Yi,[1] Qunxing An,[1] Xingbin Hu,[1] Wen Yin[1]

**ABSTRACT** Multidrug-resistant *Klebsiella pneumoniae* strains, especially carbapenem-resistant *K. pneumoniae*, have become a rapidly emerging crisis worldwide, greatly limiting current therapeutic options and posing new challenges to infection management. Therefore, it is imperative to develop novel and effective biological agents for the treatment of multidrug-resistant *K. pneumoniae* infections. Platelets play an important role in the development of inflammation and immune responses. The main component responsible for platelet antibacterial activity lies in the supernatant stimulated by gram-positive bacteria. However, little research has been conducted on the interaction of gram-negative bacteria with platelets. Therefore, we aimed to explore the bacteriostatic effect of the supernatant derived from platelet-*K. pneumoniae* coculture and the mechanism underlying this effect to further assess the potential of platelet-bacterial coculture supernatant. We conducted this study on the gram-negative bacteria *K. pneumoniae* and CRKP and detected turbidity changes in *K. pneumoniae* and CRKP cultures when grown with platelet-*K. pneumoniae* coculture supernatant added to the culture medium. We found that platelet-*K. pneumoniae* coculture supernatant significantly inhibited the growth of *K. pneumoniae* and CRKP *in vitro*. Furthermore, transfusion of platelet-*K. pneumoniae* coculture supernatant alleviated the symptoms of *K. pneumoniae* and CRKP infection in a murine model. Additionally, we observed apoptosis-like changes, such as phosphatidylserine exposure, chromosome condensation, DNA fragmentation, and overproduction of reactive oxygen species in *K. pneumoniae* following treatment with the supernatant. Our study demonstrates that the platelet-*K. pneumoniae* coculture supernatant can inhibit *K. pneumoniae* growth by inducing an apoptosis-like death, which is important for the antibacterial strategies development in the future.

**IMPORTANCE** With the widespread use of antibiotics, bacterial resistance is increasing, and a variety of multi-drug resistant Gram-negative bacteria have emerged, which brings great challenges to the treatment of infections caused by Gram-negative bacteria. Therefore, finding new strategies to inhibit Gram-negative bacteria and even multi-drug-resistant Gram-negative bacteria is crucial for treating infections caused by Gram-negative bacteria, improving the abuse of antibiotics, and maintaining the balance between bacteria and antibiotics. *K. pneumoniae* is a common clinical pathogen, and drug-resistant CRKP is increasingly difficult to cure, which brings great clinical challenges. In this study, we found that the platelet-*K. pneumoniae* coculture supernatant can inhibit *K. pneumoniae* growth by inducing an apoptosis-like death. This finding has inspired the development of future antimicrobial strategies, which are expected to improve the clinical treatment of Gram-negative bacteria and control the development of multidrug-resistant strains.

Address correspondence to Wen Yin, yinwen@fmmu.edu.cn, Xingbin Hu, hxbyqh@163.com, or Qunxing An, bestar01@163.com.

Wenting Wang, Yaozhen Chen, Yutong Chen, and Erxiong Liu contributed equally to this article. Author order was determined in the order of structures described in this article.

The authors declare no conflict of interest.

See the funding table on p. 18.

**KEYWORDS** platelet, *Klebsiella pneumoniae*, carbapenem-resistant *K. pneumoniae*, infection, reactive oxygen species, apoptosis-like death

*K*lebsiella pneumoniae is one of the common pathogens responsible for drug-resistant opportunistic infections in hospitalized patients (1), and drug-resistant strains of this bacterium are often resistant to multiple antimicrobials, a phenomenon termed "multidrug resistance" (2, 3). *K. pneumoniae*, a member of the *Enterobacteriaceae* family, is an opportunistic gram-negative pathogen that poses significant clinical and public health threats (4). In recent years, *K. pneumoniae* has emerged as one of the most common factors responsible for hospital- and community-acquired infections and a major cause of neonatal sepsis (5). Due to multidrug-resistant *K. pneumoniae* strains producing extended-spectrum β-lactamases or carbapenemases, *K. pneumoniae* spreads rapidly worldwide, making the treatment of infections caused by these strains difficult (6–8). It is reported that in Europe alone, such strains cause more than 90,000 infections and more than 7,000 deaths annually. The pathogenicity and multidrug resistance of *K. pneumoniae* make its treatment even more challenging (9). Therefore, to deal with the threat of bacterial resistance and improve the treatment of *K. pneumoniae* infections, innovative antibacterial agents and infection control strategies are urgently needed (10, 11).

A large number of studies have shown that platelets (PLTs）play an important role in host defense against pathogens (12). Platelets are anucleated cells derived from megakaryocytes in the bone marrow. Platelets not only play a role in the classical coagulation pathway but also actively participate in the immune response against pathogenic microorganisms (13). Studies have shown that platelets participate in the regulation of immune cells by releasing platelet microvesicles (PMVs), lipid mediators, nucleoside, mitochondrial DNA, growth factors, cytokines, and chemokines (13). Moreover, platelets are also directly involved in immune defense. Platelets participate directly in a bidirectional interaction with microbial pathogens, as they can sense and respond to bacterial infection signals, but at the same time, some pathogenic microorganisms can alter and modulate the structure and function of platelets (14, 15). Similar to innate immune cells, platelets contain a variety of pattern recognition receptors, such as Toll-like receptors (TLRs), C-type lectin receptors, and nucleotide-binding oligomerization domain-like receptors, which recognize different components of bacteria during infection (16, 17). Furthermore, platelets act as sentinel cells that can sense the invasion of pathogens and neutralize the invading bacteria (13). Specifically, activated platelets release antimicrobial peptides via α-granules, which perform important immune functions (12, 18). When pathogens such as bacteria or viruses induce platelet activation (19), it is mainly the α-granules that release antimicrobial peptides containing β-defensin 2, thymosin β4, and other derived antimicrobial peptides (such as fibrinopeptide A and B) to fight against the pathogens directly (20, 21). Humans have more than 30 genes coding for β-defensins, mainly expressed in epithelial cells, but they are also expressed in platelets and are secreted after thrombin stimulation (22). Most of the antibacterial effects of platelets have been shown to be associated with their secretory ability induced by bacteria, with previous reports showing platelets engulfing *S. aureus* and methicillin-resistant *S. aureus* (23). Platelet-derived transforming growth factor beta 1 plays an important role in inhibiting the growth of *S. aureus* (24). Furthermore, various antimicrobial peptides released from platelets, such as thrombocidin-1 (TC-1) and thrombocidin-2 (TC-2), have been found to exert different antibacterial effects against different bacteria. Both the components (TC-1 and TC-2) exhibit a potent bactericidal activity against *Bacillus subtilis*, to a lesser extent against *Escherichia coli* and *S. aureus,* and a fungicidal activity against *Cryptococcus neoformans* (12). Platelets contain stable messenger RNA transcripts and the translation machinery for protein synthesis despite being anucleated cells, as they inherit them from their megakaryocytic precursors (25). This special feature precisely facilitates the release of a variety of complex types of inhibitory protein molecules by the platelets.

The current research on the antibacterial mechanism of platelets mainly focuses on the direct and indirect antibacterial effects of platelets. The direct antibacterial effect is manifested as the interaction between platelet glycoproteins, particles, bacterial surface proteins, and antimicrobial peptides, such as CXCL4, CXCL7, TGF-β, and β-defensin. Indirect antibacterial effect is mediated by plasma proteins such as fibrinogen, von Willebrand factor, immunoglobulin G (IgG), platelet glycoproteins GPIIbIIIa and GPIbα, FcγRIIα, complement receptors, and TLRs (26–28).

The role of platelets in the innate immune system has been well established in previous studies. Platelets directly induce DNA damage and the inhibition of *S. aureus* division (23, 24). Platelets have also been shown to inhibit the growth of methicillin-resistant *S. aureus* by inducing hydroxyl radical-mediated apoptosis-like cell death (29). Meanwhile, the supernatant of platelets incubated with *S. aureus* has been reported to exert the same inhibitory effect as platelets do on the bacteria (30). A wealth of research on gram-positive bacteria has demonstrated that it is crucial for platelets to release bactericidal agents when they bind bacteria and get activated (31, 32). However, there is a paucity of studies on the interaction between gram-negative bacteria and platelets, and little is known about the mechanisms underlying the action of gram-negative bacteria. Moriarty et al. (31). have shown that *E. coli* can induce platelet aggregation and antibacterial effect in an IgG-dependent and FcγIIa-dependent manner, but the secretion of antimicrobial substances remains not clarified (33). In addition, due to the diversity and complexity of pathogens, the mechanism underlying platelet-mediated antibacterial defense and the clinical potential of platelets remain poorly understood.

To investigate the inhibitory effect of platelets on *K. pneumoniae* bacteria and the underlying mechanism, we assessed whether the supernatant obtained from platelet-*K. pneumoniae* coculture possesses an antibacterial activity that can inhibit *K. pneumoniae* growth. The research on the function and mechanism of antibacterial effect of platelets may pave the way for the development of novel antibacterial drugs and help in mitigating the abuse of antibiotics and controlling the spread of multidrug-resistant bacteria.

## RESULTS

### Supernatant of platelet-*K. pneumoniae* coculture inhibits *K. pneumoniae* growth *in vitro*

Using flow cytometry, we determined that the purity of washed platelets (CD41a$^+$) was more than 90% (Fig. S1A through C). Platelets (100–300 × 10$^9$/L) and *K. pneumoniae* (10$^8$ CFU/mL) were cocultured, and the supernatant of the platelet-*K. pneumoniae* coculture was obtained (Fig. S2A through C). Then, the platelet-*K. pneumoniae* coculture supernatant was extracted and added to the *K. pneumoniae* cultures, with the optimal ratio of the supernatant to *K. pneumoniae* being set to 10$^5$ CFU/mL. We observed that the addition of the platelet-*K. pneumoniae* coculture supernatant inhibited the growth of *K. pneumoniae* cultures. According to the optical density at 600 nm (OD$_{600nm}$) value and bacterial colony counts of *K. pneumoniae*, we found that with the extension of coculture time, the bacteria density of the supernatant-treated *K. pneumoniae* group was significantly lower than that of the *K. pneumoniae* group and S$_{PLT}$ + KP control group at 8 h, and the inhibition of *K. pneumoniae* growth mediated by the platelet-*K. pneumoniae* coculture supernatant lasted for 24 h (Fig. 1A and B). As shown in Fig. 1C and D, turbidity of the supernatant-treated *K. pneumoniae* group was much lower than that of the *K. pneumoniae* and S$_{PLT}$ + KP control group. In addition, the counts of bacteria in Luria-Bertani (LB) plates were markedly lower than those in *K. pneumoniae* and S$_{PLT}$ + KP control group at 8 h (Fig. 1E and F).

To explore whether the antibacterial effect of the supernatant was mainly due to platelet activation after co-culture of platelets with KP, we measured the platelet activation rate after co-culture of platelets with *K. pneumoniae*. The results showed that platelet activation was significantly increased after coculture of platelets with *K. pneumoniae* (Fig. S3A and B). In addition, when platelet activation was inhibited by the

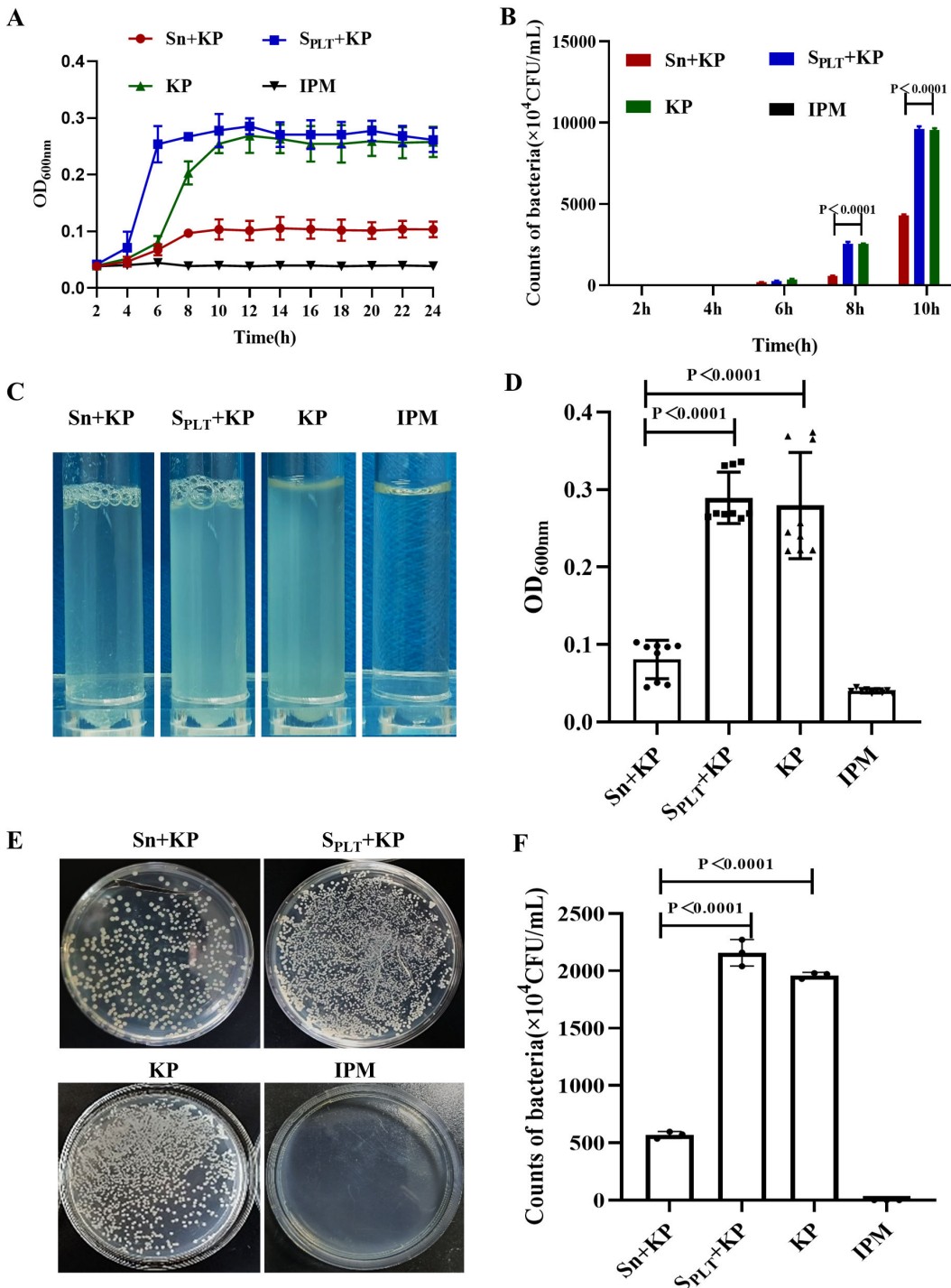

**FIG 1** Platelet-*K. pneumoniae* coculture supernatant inhibits *K. pneumoniae* growth *in vitro*. *K. pneumoniae* was added to the platelet-*K. Pneumoniae* coculture supernatant. Sn +KP: *K. pneumoniae* cocultured with platelet-*K. pneumoniae* coculture supernatant. $S_{PLT}$ + KP: *K. pneumoniae* cocultured with supernatant of platelet lysate, as a control group. KP: untreated *K. pneumoniae*, as a negative control. IPM: *K. pneumoniae* treated with Imipenem, as a positive control. (A) The growth curve of *K. pneumoniae* cocultured with or without platelet-*K. pneumoniae* coculture supernatant for 24 h, according to $OD_{600nm}$. (B) The counts of *K. pneumoniae* in each group at 2, 4, 6, 8, and 10 h. (C) The photos of *K. pneumoniae* in each group at 8 h. (D) Analysis of $OD_{600nm}$ in each group at 8 h. (E) The counts of *K. pneumoniae* in each group at 8 h. (F) Statistical results of the counts in (E). All results have been tested at least three times. Statistical analysis of the data was performed by one-way analysis of variance (ANOVA), Tukey multiple comparisons.

antiplatelet drugs Aspirin and Ticagrelor (Fig. S3C through F), the antibacterial effect of the supernatant of platelet-*K. pneumoniae* co-culture was reduced (Fig. S3G). These results demonstrate that the platelet-*K. pneumoniae* coculture supernatant can directly inhibit *K. pneumoniae* growth *in vitro* and platelet activation is the main reason for the inhibitory effect of the platelet-*K. pneumoniae* coculture supernatant.

## Platelet-*K. pneumoniae* coculture supernatant inhibits *K. pneumoniae* growth in a murine infection model

To confirm our findings *in vivo*, we aimed to replicate the inhibition of *K. pneumoniae* growth by the platelet-*K. pneumoniae* coculture supernatant in a murine infection model. First, C57BL/6 mice were infected with *K. pneumoniae* via transtracheal injection to establish a *K. pneumoniae* infection murine model (Fig. S4A). The body weight, number of white blood cells (WBCs), PLTs, and level of hemoglobin (Hb) of mice decreased after injecting *K. pneumoniae* (Fig. S4B through E). Moreover, the number of *K. pneumoniae* in the lungs and bronchoalveolar lavage fluid (BALF) of *K. pneumoniae*-infected mouse group was markedly higher than that in the PBS-treated mouse group (Fig. S4F through H). The mice were then euthanized 24 h after infection, and the mouse lungs were analyzed using hematoxylin and eosin staining. The results showed that there were more leukocytes in the lungs of the *K. pneumoniae*-infected group, and the lung histological injury score (34, 35) was higher than that of the control group (Fig. S4I and J). These data suggested that the *K. pneumoniae* infection murine model was successfully established in this study.

To determine whether the administration of platelet-*K. pneumoniae* coculture supernatant inhibits *K. pneumoniae* growth and relieves *K. pneumoniae* infection in mice, the supernatant obtained from the platelet-*K. pneumoniae* coculture was transfused into the *K. pneumoniae*-infected mice after 24 h infection (Fig. 2A). We observed the changes in body weight, WBCs, PLTs, and Hb in mice within 48 h (Fig. S5A through D) and found that the infection symptoms in mice were alleviated following treatment with the platelet-*K. pneumoniae* coculture supernatant. The body weight of mice in the *K. pneumoniae*-infected and supernatant-treated groups was lower than that in the PBS group (Fig. S5E). The counts of WBCs in the *K. pneumoniae*-infected and supernatant-treated groups significantly decreased compared to that in the PBS group (Fig. S5F). Meanwhile, PLTs count in the supernatant-treated group was lower than that in the PBS and *K. pneumoniae* group (Fig. S5G). Finally, there was no difference in the levels of Hb across the three groups (Fig. S5H). The mice were euthanized 48 h after *K. pneumoniae* infection, and we observed that the entire lungs of all mice from the *K. pneumoniae*-infected mouse group exhibited red appearance, indicating a significant bleeding in the lungs. However, lung bleeding was alleviated after treatment with the platelet-*K. pneumoniae* coculture supernatant (Fig. 2B). Notably, the number of *K. pneumoniae* in the lungs and BALF was significantly reduced after 48 h infection (Fig. 2C and D; Fig. S5I). Additionally, damage to the lungs was also reduced in mice from the supernatant-treated group, as assessed from pathological examination (Fig. 2E and F). Moreover, the analysis of BALF cells using Swiss Giemsa staining showed significant changes in the WBC and red blood cell (RBC) counts, indicating that the inflammatory cell infiltration and hemorrhage in the mouse lungs were alleviated (Fig. 2G through I). Furthermore, the cytokine levels in BALF indicated that the release of proinflammatory cytokines in the supernatant-treated mouse group was reduced compared with that in the *K. pneumoniae* group (Fig. 2J through M). These results suggest that the platelet-*K. pneumoniae* coculture supernatant can inhibit *K. pneumoniae* growth *in vivo*.

## Platelet-*K. pneumoniae* coculture supernatant induces oxidative stress damage and apoptosis-like death in *K. pneumoniae*

As the platelet-*K. pneumoniae* coculture supernatant was found to exert inhibitory effects on *K. pneumoniae in vivo* and *in vitro,* we sought to reveal the potential mechanism underlying this process. To this end, we first used scanning electron microscopy to

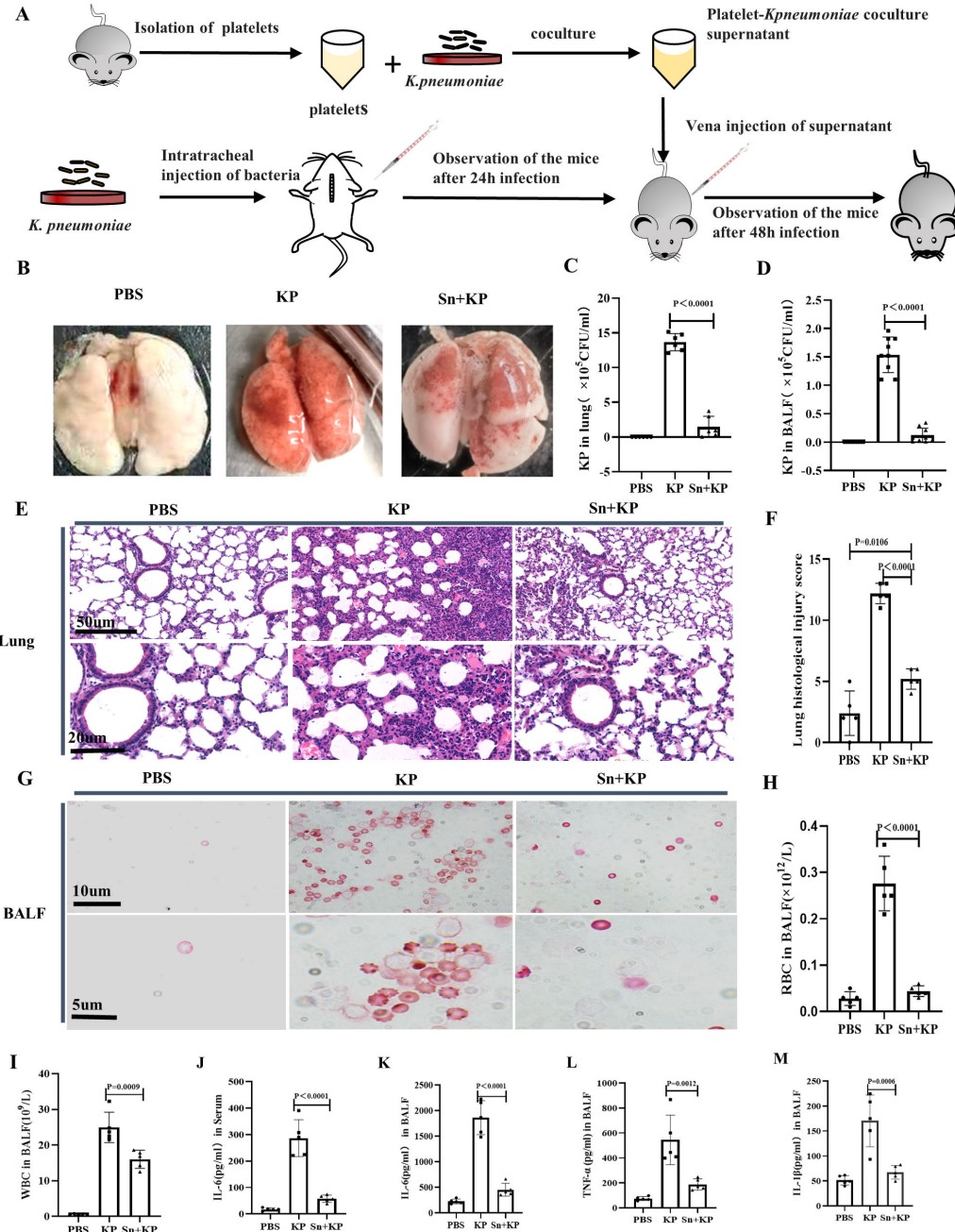

**FIG 2** Platelet-*K. pneumoniae* coculture supernatant transfusion alleviates the pulmonary symptoms of *K. pneumoniae*-infected mice. C57BL/6 mice were infected with *K. pneumoniae* by intratracheal injection and then treated with or without platelet-*K. pneumoniae* coculture supernatant (*n* = 5). PBS: mice were injected intratracheal with PBS as a negative control group. KP: mice were injected intratracheal with *K. pneumoniae*. Sn +KP: mice were injected with platelet-*K. pneumoniae* coculture supernatant after *K. pneumoniae* infection. (A) Schematic diagram of platelet-*K. pneumoniae* coculture supernatant transfusion in *K. pneumoniae* infected mice. (B) Lungs of mice in each group. (C) Statistical histogram of bacteria count in lung. (D) Statistical histogram of bacteria count in BALF. (E) Representative images of H&E staining of lung. (F) Statistical histogram of histological score analysis of lung injury in the mice in (E). (G) Representative images of Swiss staining analysis of cell precipitates in BALF. (H) Statistical histogram of the RBC count of BALF in each group. (I) Statistical histogram of the WBC count of BALF in each group. (J) Statistical histogram of IL-6 level in the serum of the mice in each group. (K) Statistical histogram of IL-6 level in the BALF of the mice in each group. (L) Statistical histogram of TNF-α level in the BALF of the mice in each group. (M) Statistical histogram of IL-1β level in the BALF of the mice in each group. All results have been tested at least three times. Statistical analysis of the data was performed by one-way ANOVA, Tukey multiple comparisons.

observe the ultrastructure of the bacterium. When *K. pneumoniae* was cultured with the platelet-*K. pneumoniae* coculture supernatant in the culture medium, its morphology became significantly distorted and shrunken (Fig. 3A). Moreover, as the formation of biofilm is part of the dynamic process of bacterial growth and biofilm can enhance the resistance of bacteria to host defense and antibiotics (36–38), we used scanning electron microscopy to observe biofilm formation in the *K. pneumoniae* cultures. We found that biofilm formation was significantly reduced in the supernatant-treated group compared with that in the *K. pneumoniae* control group (Fig. 3B).

Our previous study has shown that platelets can induce DNA damage in *S. aureus* and methicillin-resistant *S. aureus*. Therefore, we hypothesized that the supernatant of platelet-*K. pneumoniae* coculture may also inflict DNA damage in *K. pneumoniae*. Indeed, we observed changes in the morphology and internal structure of *K. pneumoniae* using transmission electron microscopy after the treatment of *K. pneumoniae* cultures with the platelet-*K. pneumoniae* coculture supernatant. As shown in Fig. 3C, we observed an evident agglutination of *K. pneumoniae* chromatin as well as irregular eversion and blurring of the cell wall and membrane after treatment with the platelet-*K. pneumoniae* coculture supernatant. These morphological outcomes showed that the cell wall, cell membrane, and cell nucleus of *K. pneumoniae* were damaged following supernatant treatment.

Antibiotic treatment can induce cell apoptosis and cause the targeted cells to exhibit characteristic markers of apoptosis, including phosphatidylserine (PS) exposure, chromosome condensation, and DNA fragmentation (29, 39). We found that the platelet-*K. pneumoniae* coculture supernatant had an antibiotic-like effect on *K. pneumoniae* in our previous study. Therefore, PS exposure in *K. pneumoniae* was assessed using Annexin V staining and flow cytometry, which confirmed the occurrence of apoptosis-like death in *K. pneumoniae* following treatment with the platelet-*K. pneumoniae* coculture supernatant. The percentages of Annexin V-positive *K. pneumoniae* in the supernatant-treated and control groups were 6.91% and 1.14%, respectively (Fig. 3D and E), suggesting that the platelet-*K. pneumoniae* coculture supernatant treatment can induce PS exposure in *K. pneumoniae*.

Next, the membrane potential of supernatant-treated *K. pneumoniae* was measured using the lipophilic anionic fluorescent dye DiBAC$_4$(3), which is sensitive to cellular membrane potential. Compared to that in the untreated *K. pneumoniae* group, the fluorescence intensity of DiBAC$_4$(3) in supernatant treatment group increased significantly (Fig. 3F and G), indicating that treatment with the supernatant induced depolarization of the *K. pneumoniae* membrane potential. We also assessed the activity of intracellular caspase in *K. pneumoniae* using V-A-D-FMK. Intracellular fluorescence indicated a stable binding of V-A-D-FMK-FITC to bacterial proteins that possess binding affinity for a general caspase substrate. The increase in fluorescence intensity reflects an increase in the concentration of these proteins. Results showed that the fluorescence intensity of V-A-D-FMK was enhanced in the supernatant-treated group, implying that the activity of intracellular caspase increased in *K. pneumoniae* after supernatant treatment (Fig. 3H and I).

DNA fragmentation is one of the most important indicators of apoptosis (39). In the previous morphological observations, we could see that the platelet-*K. pneumoniae* coculture supernatant caused chromatin condensation in *K. pneumoniae*; hence, we sought to evaluate DNA damage in *K. pneumoniae* using the terminal deoxynucleotidyl transferase dUTP nick-end labeling (TUNEL) assay and flow cytometry. The fluorescence intensity of the labeled DNA fragments increased significantly after treatment with the platelet-*K. pneumoniae* coculture supernatant, further validating that DNA damage, indeed, occurred in *K. pneumoniae* after treatment with the supernatant (Fig. 3J and K).

Reactive oxygen species (ROS) are a product of biological respiration metabolism. When cells are exposed to external environmental pressures, oxidative stress occurs and excessive ROS are produced, adversely damaging the DNA and proteins in cells, eventually leading to cell death (40). Therefore, we assessed the levels of ROS using

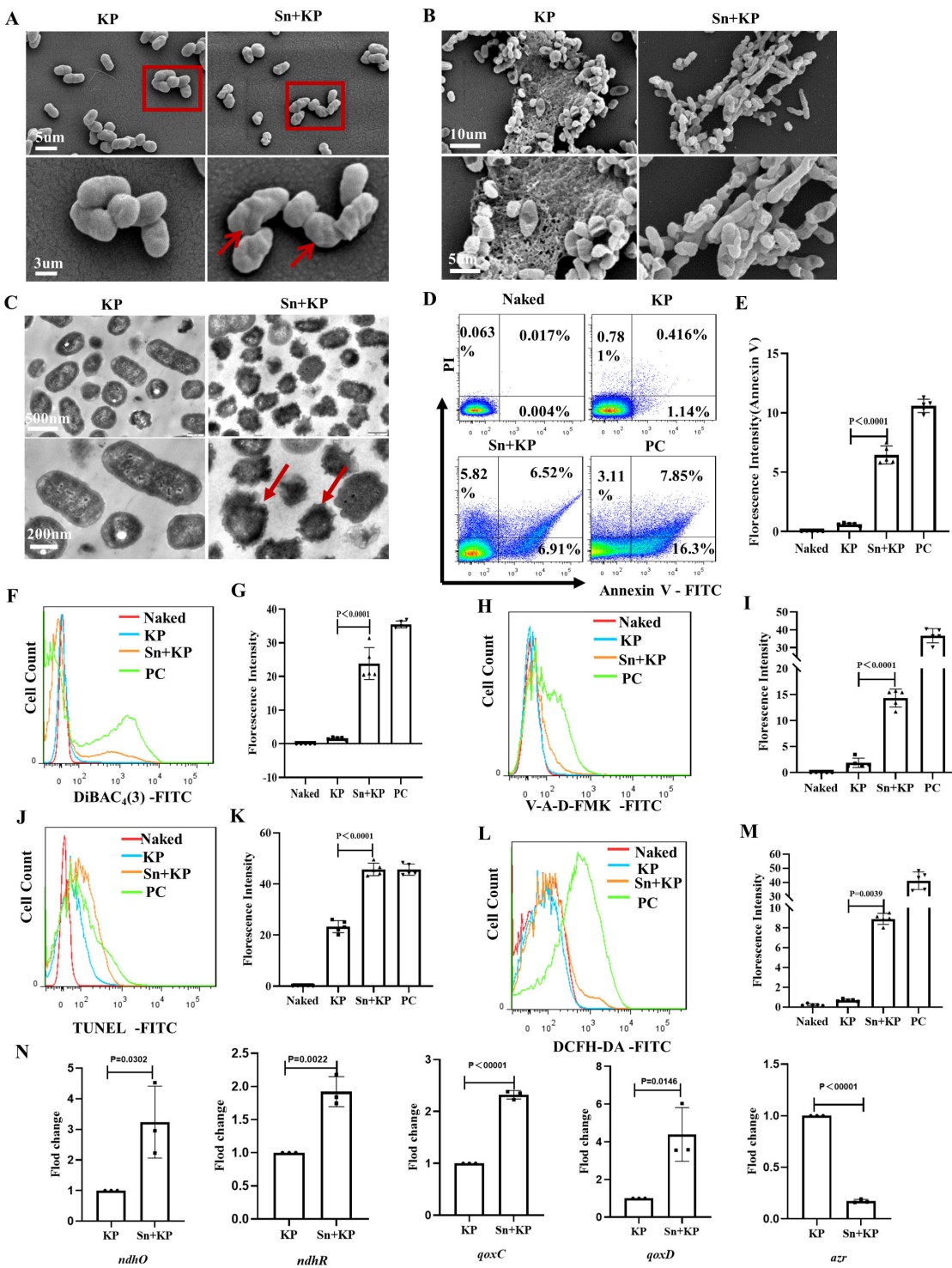

**FIG 3** Platelet-*K. pneumoniae* coculture supernatant induces apoptosis-like death of *K. pneumoniae*. *K. pneumoniae* were cocultured with or without platelet-*K. pneumoniae* coculture supernatant for 6 h. (A and B) The ultrastructure of *K. pneumoniae* was observed by scanning electron microscopy. (A) Scale bars: 5 µm (upper panel); 3 µm (lower panel). The original magnification was 10,000 (upper panel), 20,000 (lower panel) in each group. (B) Scale bars: 10 µm (upper panel); 5 µm (lower panel). The original magnification was 5,000 (upper panel), 10,000 (lower panel) in each group. (C) The ultrastructure of *K. pneumoniae* was observed by transmission electron microscopy. Scale bar: 500 nm (upper panel); 200 nm (lower panel). The original magnification was 20,000 (upper panel), 40,000 (lower panel) in each group. (D) PS exposure was detected by flow cytometry. (E) Statistical results of FITC-Annexin V fluorescence intensity in (D). (F) DiBAC$_4$(3)-labeled bacterial membrane potential was detected by flow cytometry. (G) Statistical results of FITC-DiBAC$_4$(3) fluorescence intensity in (F). (H) V-A-D-FMK labeled intracellular caspase of bacteria was detected by FACS analysis. (I) Statistical results of V-A-D-FMK fluorescence intensity in (H). (J) DNA damage of *K. pneumoniae*

**FIG 3** (Continued)

was detected by FACS analysis with TUNEL. (K) Statistical results of fluorescence intensity in (J). (L) FITC-DCFH-DA labeled intracellular ROS of *K. pneumoniae* was detected by flow cytometry. (M) Statistical results of fluorescence intensity in (L). (N) Validation of RNA-sequence results by qRT-PCR. Naked: unstained *K. pneumoniae*. KP: untreated *K. pneumoniae*, as a negative control. Sn +KP: *K. pneumoniae* treated with platelet-*K. pneumoniae* coculture supernatant. PC: *K. pneumoniae* treated with MMC (5 µg/mL) for 6 h, as a positive control. All results have been tested at least three times. Statistical analysis of the data was performed by one-way ANOVA, Tukey multiple comparisons.

DCFH-DA-labeled probe and found that the fluorescence intensity of DCFH-DA in *K. pneumoniae* increased after treatment with the platelet-*K. pneumoniae* coculture supernatant (Fig. 3L and M). We detected the expression of ROS-related stress genes in *K. pneumoniae* and found that the expression of *ndhO*, *ndhR*, *qoxC*, and *qoxD* genes was upregulated and the *azr* gene was downregulated after platelet-*K. pneumoniae* coculture supernatant treatment (Fig. 3N). Increased expression of *qoxC* and *qoxD* suggests reduced electron supply to the oxidative respiratory chain, and the decreased expression of AZR indicates compromised electron supply to the oxidative respiratory chain (29, 41–43) .These results provide evidence that *K. pneumoniae* suffered oxidative stress following platelet-*K. pneumoniae* coculture supernatant treatment.

Additionally, we examined the levels of pH, lactate dehydrogenase (LDH), glucose, and other ions in *K. pneumoniae* cultures treated with the platelet-*K. pneumoniae* coculture supernatant to explore whether the changes in physical or chemical environments affect the growth of *K. pneumoniae* (Fig. S6A through G). Results showed that the levels of LDH, $Na^+$, $Cl^-$, $CO_2$, and pH increased, while those of $K^+$ and $Ca^{2+}$ decreased (Fig. S6H through N). These data indicated that the osmotic pressure in *K. pneumoniae* was altered following treatment with the platelet-*K. pneumoniae* coculture supernatant, which might partially explain the damage to the cell membrane and cell wall. Meanwhile, the increase in LDH levels further indicated that the metabolism of *K. pneumoniae* was gravely affected. Collectively, these data suggested that the platelet-*K. pneumoniae* coculture supernatant induced oxidative stress damage and apoptosis-like changes in *K. pneumoniae*. The imbalance of $Na^+$, $Cl^-$, $K^+$, and $Ca^{2+}$ concentrations and the increase in LDH levels also indicated that the homeostasis and metabolic process in *K. pneumoniae* were adversely impaired, potentially causing injury and death of the bacterial population.

## *N*-acetylcysteine inhibits the oxidative stress injury and apoptosis-like death induced upon treatment with the platelet-*K. pneumoniae* coculture supernatant

We have demonstrated that the production of intracellular ROS increased and the apoptosis-like changes appeared following treatment of *K. pneumoniae* with the platelet-*K. pneumoniae* coculture supernatant. We hypothesized that apoptosis was triggered by the overproduction of ROS induced by antimicrobial peptides in coculture supernatant.

To confirm whether the increased ROS levels are related to the apoptosis-like death of *K. pneumoniae* after treatment with the platelet-*K. pneumoniae* coculture supernatant, the antioxidant *N*-acetylcysteine (NAC) was added into the coculture system, and the changes in PS exposure, membrane potential, and intracellular caspase activity of *K. pneumoniae* were assessed. After the production of ROS in the supernatant-treated *K. pneumoniae* cultures was inhibited by NAC (Fig. 4G and H), we discovered that the decline in ROS levels led to an increase in *K. pneumoniae* population in the cultures (Fig. S7A through C) as well as reduction in the PS exposure (Fig. 4A and B), cell membrane depolarization (Fig. 4C and D), and intracellular caspase activity (Fig. 4E and F) of *K. pneumoniae*. These results signified that the apoptosis-like changes in *K. pneumoniae* were attenuated because ROS production was inhibited. Taken together, the platelet-*K. pneumoniae* coculture supernatant appeared to induce apoptosis-like death in *K. pneumoniae* by promoting the overproduction of ROS (Fig. S7D).

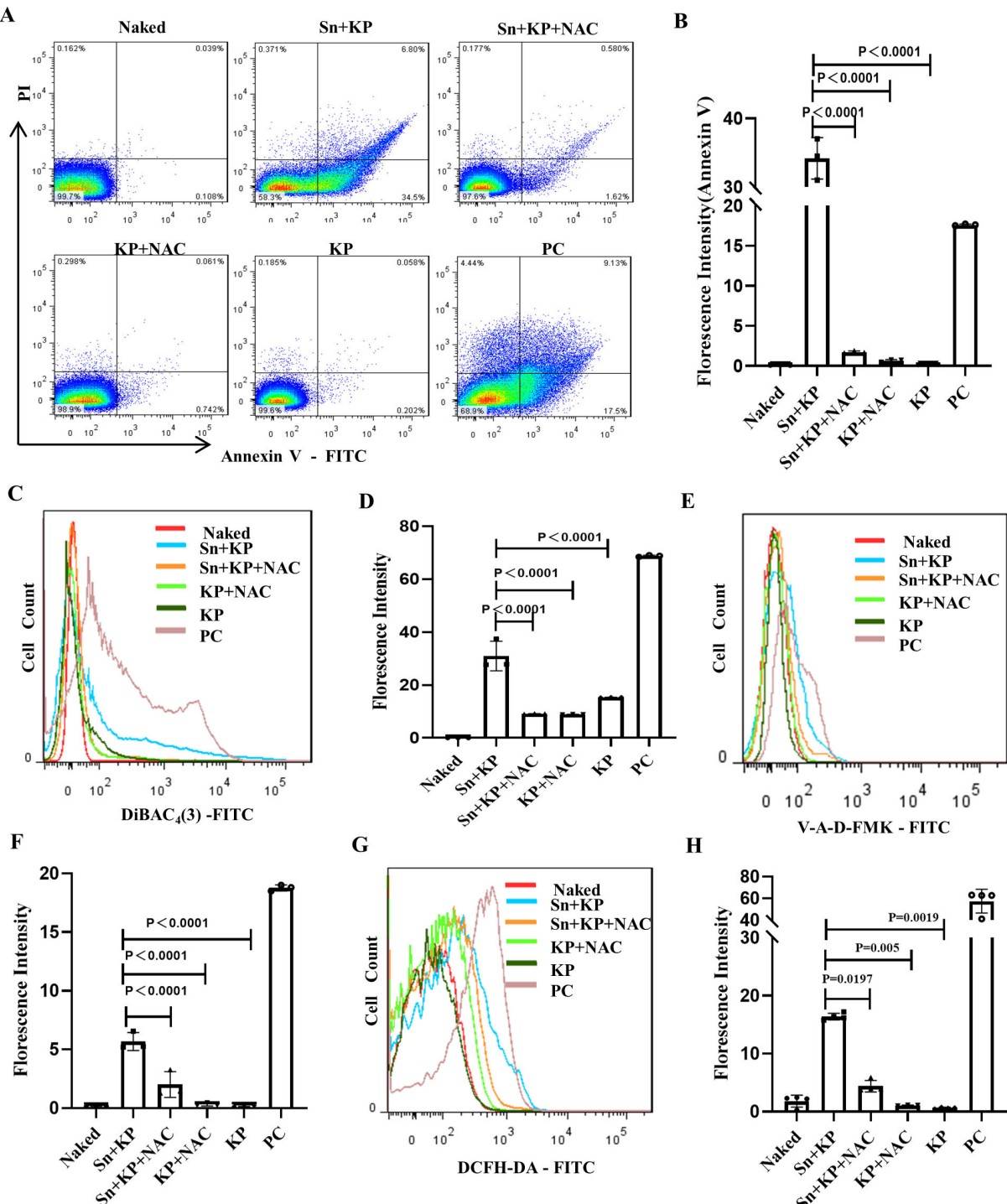

**FIG 4** NAC inhibits the apoptosis of *K. pneumoniae* induced by platelet-*K. pneumoniae* coculture supernatant. *K. pneumoniae* were cocultured with or without platelet-*K. pneumoniae* coculture supernatant for 8 h, and 6 mM NAC was added to the coculture system. Naked: unstained *K. pneumoniae*. Sn +KP: *K. pneumoniae* treated with platelet-*K. pneumoniae* coculture supernatant. Sn +KP + NAC: NAC was added to the platelet-*K. pneumoniae* coculture supernatant cocultured with *K. pneumoniae*. KP + NAC: NAC was added to 1640 RPMI medium with *K. pneumoniae*. KP: untreated *K. pneumoniae*, as a negative control. PC: *K. pneumoniae* treated with MMC (5 µg/mL) for 6 h, as a positive control. (A) PS exposure was detected by flow cytometry. (B) Statistical results of FITC-Annexin V fluorescence intensity in (A). (C) Bacterial membrane potential of *K. pneumoniae* was detected by flow cytometry with $DiBAC_4(3)$. (D) Statistical results of $DiBAC_4(3)$ fluorescence intensity in (C). (E) Intracellular caspase of *K. pneumoniae* was detected by flow cytometry with V-A-D-FMK. (F) Statistical results of V-A-D-FMK fluorescence intensity in (E). (G) The ROS of *K. pneumoniae* was detected by flow cytometry with DCFH-DA.(H) Statistical results of DCFH-DA fluorescence intensity in (G). All results have been tested at least three times. Statistical analysis of the data was performed by one-way ANOVA, Tukey multiple comparisons.

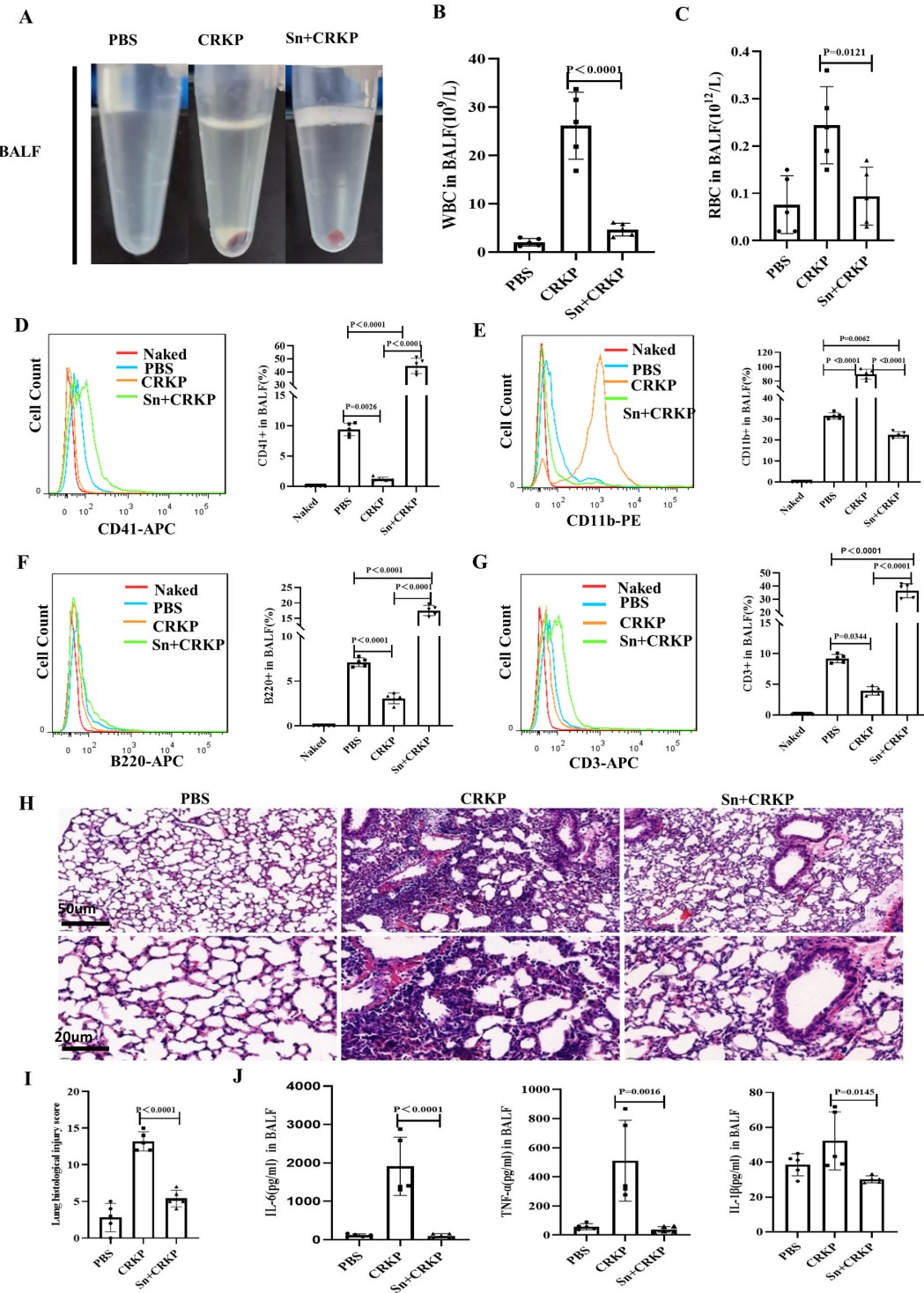

**FIG 5** Platelet-*K. pneumoniae* coculture supernatant transfusion alleviates the the pulmonary symptoms of CRKP infected mice. C57BL/6 mice were infected with CRKP by intratracheal injection and then treated with or without platelet-*K. pneumoniae* coculture supernatant(*n* = 5). PBS: mice were injected with PBS into the trachea, as negative control group. CRKP: mice were injected with CRKP into the trachea. Sn+CRKP: mice were injected with platelet-*K. pneumoniae* coculture supernatant after CRKP infection. (A) Photos of cells in BALF in each group. (B) Statistical histogram of the WBC count in BALF of each group. (C) Statistical histogram of the RBC count in BALF of each group. (D) The detection of the percentage of CD41 positive cells by flow cytometry. (E) The detection of the percentage of CD11b positive cells by flow cytometry. (F) The detection of the percentage of B220 positive cells by flow cytometry. (G) The detection of the percentage of CD3 positive cells by flow cytometry. (H) Representative images of H&E staining of lung. (I) Statistical histogram of histological score analysis of lung injury in the mice in (H). (J) Statistical histogram of IL-6, TNF-α, and IL-1β levels in the BALF of the mice in each group. All results have been tested at least three times. Statistical analysis of the data was performed by one-way ANOVA, Tukey multiple comparisons.

## Platelet-*K. pneumoniae* coculture supernatant exerts antimicrobial effects on CRKP *in vitro* and *in vivo*

To further explore whether the platelet-*K. pneumoniae* coculture supernatant exerts the same direct antibacterial effects on CRKP, we monitored the growth of CRKP after platelet-*K. pneumoniae* coculture supernatant treatment. First, clinical isolate strains were obtained, and the drug resistance and virulence genes of CRKP were determined. The results showed that the string test of CRKP was negative, the capsular serotype was *K20,* and the virulence genes were *magA*, *fimH,* and *Aero* (Fig. S9A through F). Then, the platelet-*K. pneumoniae* coculture supernatant was extracted and added to the CRKP cultures. The turbidity of the supernatant-treated CRKP group was found to be lower than that of the CRKP control group (Fig. S8A). In addition, the $OD_{600nm}$ decreased following treatment with the platelet-*K. pneumoniae* coculture supernatant (Fig. S8B). The counts of CRKP populations in LB plates were also lower than those in the control group (Fig. S8C). These observations demonstrated that the platelet-*K. pneumoniae* coculture supernatant can directly inhibit CRKP growth *in vitro.*

To verify whether platelet-*K. pneumoniae* coculture supernatant transfusion can inhibit CRKP growth and relieve CRKP infection in mice, a CRKP-infected murine model was first established via the transtracheal injection of CRKP into mice. The platelet-*K. pneumoniae* coculture supernatant was then injected into the CRKP-infected mice via the tail veins. We monitored the changes in body weight, WBC counts, PLT counts, and Hb level in mice within 48 h (Fig. S10A through G) and found that the infection symptoms in the mice were alleviated after transfusion of the supernatant. We also observed pulmonary congestion and edema in mice following euthanasia 48 h after infection. The results showed that pulmonary congestion and edema in the mice were relieved (Fig. S10H), and the number of bacterial cells in the lungs and BALF was significantly reduced after supernatant treatment (Fig. S10I). We collected and centrifuged the BALF to evaluate the cell deposits at the bottom of the tubes and found that the deposits in the CRKP-infected group contained red blood cells, with the number of cells in the supernatant-treated group being slightly lower than that in the infected group (Fig. 5A). Moreover, the analysis of BALF cells in the WBC and RBC counts, indicating that the levels of inflammatory cell infiltration and hemorrhage in the mouse lungs were alleviated after supernatant transfusion (Fig. 5B and C). The proportions of granulocytes, T lymphocytes, and B lymphocytes in the cell deposits were then analyzed via flow cytometry; results showed that the proportion of $CD11b^+$ cells in the CRKP group increased significantly, while the proportions of $CD41^+$, $CD3^+$, and $B220^+$ cells in the supernatant-treated group were much higher than those in the CRKP group (Fig. 5D through G). These results indicate that the platelet-*K. pneumoniae* coculture supernatant might play an important role in mobilizing murine immune cells against CRKP infection. In addition, hematoxylin and eosin staining showed that inflammatory cell infiltration was reduced and that the pathological damage of alveolar tissue was improved after supernatant treatment (Fig. 5H and I). In the supernatant-treated group, the levels of proinflammatory cytokines (IL-6, TNF-α, and IL-1β) were lower than those in the CRKP-infected group (Fig. 5J). Taken together, these results demonstrate that platelet-*K. pneumoniae* coculture supernatant transfusion can inhibit CRKP growth and defend against CRKP infection *in vivo.*

## DISCUSSION

In the present study, we successfully demonstrated that the platelet-*K. pneumoniae* coculture supernatant can inhibit the growth of *K. pneumoniae* both *in vivo* and *in vitro.* These findings reveal a novel role of platelets in combating the infections caused by *K. pneumoniae* and its drug-resistant strain, CRKP. In addition, this study lays a foundation for future studies on the transfusion of platelets or their derivatives and the development and application of new antibacterial agents.

Platelets are complex cells containing three different types of granules: the α, dense or δ, and lysosomal granules. The α-granules carry diverse proteins, cytokines,

chemokines, and growth factors, while the δ-granules contain small molecules such as adenosine diphosphate, serotonin, glutamate, histamine, and calcium, which are necessary for hemostasis (44). In the present study, we found that the changes in bacterial turbidity in the cultures of *K. pneumoniae* treated with platelet-*K. pneumoniae* coculture supernatant were significantly different from the changes caused by platelet-*S. aureus* coculture supernatant in the corresponding *S. aureus* cultures. Therefore, we hypothesized that antimicrobial proteins may have formed by the platelets against *K. pneumoniae* in the platelet-*K. pneumoniae* coculture supernatant and the production of these antimicrobial peptides depended on platelets activation and it may be closely related to the special translation mode of platelets (45, 46). This study clearly demonstrated that treatment with the platelet-*K. pneumoniae* coculture supernatant had a significant inhibitory effect on *K. pneumoniae in vitro* and may play an important role in ameliorating *K. pneumoniae*-induced bacterial pneumonia.

In this study, treatment with the platelet-*K. pneumoniae* coculture supernatant significantly increased ROS production and induced apoptosis-like death in *K. pneumoniae*. The inhibition induced by the platelet-*K. pneumoniae* coculture supernatant in *K. pneumoniae* cultures observed in the present study might be the combined outcome of bacterial cell membrane damage and DNA damage. In the process of mediating apoptosis-like changes in bacteria, ROS production serves not only as one of the apoptotic indicators but also as a key factor in the occurrence of apoptosis (47). A number of studies have confirmed that bacteria produce excessive ROS to interfere with the bacterial oxidative phosphorylation and tricarboxylic acid cycle (48), especially important regulators of highly harmful apoptosis, which can directly damage DNA, lipids, and proteins (33). In addition, excessive ROS can cause cell membrane depolarization, eventually leading to apoptosis-like death in bacteria (39, 47). As mentioned previously, platelet-derived antimicrobial peptides present in the platelet-*K. pneumoniae* coculture supernatant may play an important role in inhibiting *K. pneumoniae* growth, changing the ionic environment of bacterial survival and causing oxidative stress damage in bacteria (Fig. S6). Previous reports have also revealed that antimicrobial peptides, such as cationic host defense peptides, contain positive charges and can bind to anionic bacterial membranes, causing lipid clustering and increasing membrane permeabilization (45, 46), thereby inhibiting bacterial proliferation. In addition, combined treatment with NAC inhibits the antibacterial activities of daunomycin, moxifloxacin, and oxacillin against *S. aureus* (29, 48). Our study supports the hypothesis that ROS play a key role in the supernatant-mediated apoptosis-like changes in *K. pneumoniae*. The inhibitory effect of the platelet-*K. pneumoniae* coculture supernatant on *K. pneumoniae* is particularly similar to the effect of antibiotics (39, 49). More importantly, the platelet-*K. pneumoniae* coculture supernatant also exerts the same inhibitory effect on CRKP *in vivo* and *in vitro*. In addition, transfusion of platelet-*K. pneumoniae* coculture supernatant alleviated the symptoms of pneumonia in *K. pneumoniae*- and CRKP-infected mice. These findings are of great significance for the research and development of antimicrobial agents in the future and would provide novel insights into potential clinical treatment of patients with refractory or drug-resistant infections.

Although our study provides evidence that the supernatant of platelet-*K. pneumoniae* coculture mediates the inhibition of *K. pneumoniae* growth, it has several limitations. Although we noted that the main molecules that are released from platelets upon their activation (including β-defensin 2, thymosin β4, CXCL4, CXCL7, CXCL12, and TGF-β, all of which have potent antibacterial effects) have already been reported (50–54), the exact composition of the platelet-*K. pneumoniae* coculture supernatant and the components that were associated with antibacterial activity in our study are yet to be elucidated. Additionally, although the transfusion of platelet-*K. pneumoniae* coculture supernatant alleviated infection symptoms in the *K. pneumoniae*-infected mice model, our observations do not preclude the participation of neutrophils and other immune cells in producing the observed outcome *in vivo*. Despite the key role that the supernatant plays in the anti-*K. pneumoniae* effect and the induction of ROS overproduction,

the detailed mechanism underlying the platelet-*K. pneumoniae* coculture supernatant-induced apoptosis-like changes in *K. pneumoniae* remains unclear, and this is a research direction we would like to undertake in the future.

In conclusion, Our results demonstrated that the platelet-*K. pneumoniae* coculture supernatant directly inhibited the growth of *K. pneumoniae* and induced apoptosis-like death in *K. pneumoniae.* Additionally, we showed the overproduction of ROS-mediated oxidative stress damage in *K. pneumoniae*. The inhibitory effect of the platelet-*K. pneumoniae* coculture supernatant on drug-resistant strain CRKP was also shown in our study. As platelets are related to infection and inflammation, the antimicrobial function that platelets may perform cannot be ignored, but more research is needed to determine whether the platelet-bacteria coculture supernatant exhibits similar antimicrobial effects even in the presence of other pathogenic microorganisms in such a coculture setup. It is imperative that our understanding of the functions of platelets and platelet-related derivatives be improved, as this may provide alternative approaches for improving antimicrobial therapy, developing novel antimicrobial agents, controlling bacterial diseases, and solving the emerging crisis of antibiotic resistance.

## MATERIALS AND METHODS

### Preparation of human washed platelets

Human apheresis platelets were collected from 8 donors, centrifuged at 213$g$ for 5 min, and then washed gently with washing solution (PBS containing 10% ACD solution) twice. Residual plasma and red blood cells were removed after washing, and the platelets were resuspended to achieve a count of $200 \times 10^9$/L in 1640 RPMI medium (Gibco). The purity of the washed platelets was detected by blood autoanalyzer (XP-100, Sysmex, Kobe, Japan) and flow cytometry. All relevant procedures were performed in accordance with the requirements of the Xijing Hospital Ethics Committee (KY20212120-C-1).

### Preparation of mice washed platelets

C57BL/6 mice, 6- to 8-week-old, were from the Animal Center of the Fourth Military Medical University. All animal experiments were conducted in accordance with the regulations on animal protection and management of the Fourth Military Medical University (IACUC-20130014). In this study, blood was collected from the hearts of mice and stored in PBS solution (contained 10% ACD solution). The blood was centrifuged twice (150$g$ each time) for 5 min in order to remove the white and red blood cells. The upper layer of the fluid contained platelets, which are then transferred to a new sterile tube and washed twice. Finally, platelet precipitates were collected and suspended in 1640 RPMI medium with a count of $200 \times 10^9$ /L.

### Preparation, culture, and counting of bacteria

*K. pneumoniae* (ATCC number: 700603, Manassas, VA, USA) was inoculated on LB agar plates at 37°C for 24 h in order to obtain the single colony. Then, the single colony of *K. pneumoniae* was inoculated in LB and cultured at 37°C for 6–8 h to logarithmic growth phase.

*In vitro* experiments, *K. pneumoniae* was washed with sterile PBS for three times, and the concentration of bacteria was determined by a spectrophotometer (UV-2550; Shimadzu, Kyoto, Japan) at 600 nm wavelength ($OD_{600nm}$). Finally, *K. pneumoniae* was diluted to $1 \times 10^5$ CFU/mL for testing. The numbers of bacteria were determined through serial dilution and plating assay. Five tubes were prepared, and 900 µL sterile PBS was added to each tube. Then, added 100 µL of the bacterial solution ($1 \times 10^9$ CFU/mL) to tube 1, mixed it and absorbed 100 µL into tube 2, diluted it to tube 5 in turn. One hundred microliters of bacterial solution was for plating, and the bacteria were incubated at 37°C for 20 h. *In vivo* experiments, *K. pneumoniae* was washed with sterile

PBS three times to prevent aggregation and then suspended with PBS to the count of $1 \times 10^8$ CFU/mL.

CRKP were isolated from the clinical samples in laboratory department of Xijing Hospital of Fourth Military Medical University. CRKP was cultured and counted in the same way as *K. pneumoniae in vitro* studies, and it was diluted to $1 \times 10^5$ CFU/mL for testing. *In vivo* studies, CRKP was washed with sterile PBS three times to prevent aggregation and then suspended with PBS to the count of $1 \times 10^8$ CFU/mL. The drug resistance of clinically isolated CRKP was detected by disc diffusion test, and the presence of drug resistance genes and virulence factors was determined by PCR. In addition, tigecycline and imipenem were used to confirm drug resistance of the CRKP strains. The MICs of tigecycline for CRKP clinical isolated strains were detected by broth microdilution method in Mueller-Hinton (MH) medium. First, 180 µL of diluted bacterial solution ($1 \times 10^5$ CFU/mL) was added to the tube 1, 100 µL of diluted bacterial solution was added from tube 2 to 11, and sterile MH medium was added to the tube 12 as a control. Then, added 20 µL of the drug solution to tube 1, mixed it and absorbed 100 µL into tube 2, diluted it to tube 11 in turn, respectively. Finally, the drug concentration was 1,280, 640, 320, 160, 80, 40, 20, 10, 5, 2.5, and 1.25 µg/mL. The bacteria were incubated at 37°C for 20 h, and MIC was the lowest drug concentration to completely inhibit the growth of bacteria.

## Preparation of platelet-*K. pneumoniae* coculture supernatant

*In vitro* and *vivo* study, *K. pneumoniae* ($1 \times 10^8$ CFU/mL) and human (or mice) platelets ($200 \times 10^9$ /L) were mixed and cocultured in 1,640 RPMI medium at 37°C for 6–8 h under shaker. The coculture of *K. pneumoniae* and platelets was centrifuged at 10,000*g* for 10 min after 6–8 h of incubation. The supernatant of the coculture was then transferred to a sterile tube, and the supernatant containing secreted platelets compounds was filtered using a sterile filter (0.2 µm pore size) to eliminate all cells. The supernatant was incubated again with the same bacteria for 6 h at 37°C under shaker.

## Mice model of *K. pneumoniae* and CRKP infection

*K. pneumoniae* or CRKP (100 µL, $1 \times 10^8$ CFU/mL) was injected into the trachea of mice, and the blood was collected through the orbital veins at 0 and 24 h in each group. The volume of the blood collected each time was 30 µL, and it was stored in a PBS solution (containing 20% ACD solution) for analysis of WBC, and platelet counts using the blood autoanalyzer and the number of bacteria in BALF and lungs was counted using LB plates. The mice were euthanized 24 h after infection, and the lungs were collected for the analysis by HE staining.

## Transfusion of platelet-*K. pneumoniae* coculture supernatant into mice

C57BL/6 mice (6- to 8-week-old) were divided into PBS, KP/CRKP, and supernatant transfusion (Sn + KP/Sn + CRKP) groups (5 mice each group, 3 female, and 2 male). In the supernatant transfusion group, platelet-*K. pneumoniae* coculture supernatants (300 µL) were transfused via tail vein injection after *K. pneumoniae*/CRKP infection for 24 h, and the blood was collected through the vein at 0, 24, and 48 h. WBCs and PLTs in the peripheral blood of mice were collected at different times and detected by the blood autoanalyzer.

The mice were euthanized 48 h after *K. pneumoniae*/CRKP infection, and the needle of a syringe was inserted into the trachea of each mouse; we instilled PBS into the mouse lung and gently drew back fluid after all lobes of lung are inflated. Re-instilled the returned fluid into mouse lung and collected the first 1 mL return and put on ice, repeated at least for three times. The lungs were collected for HE staining. In each tissue sample, five random areas were scored, and the mean value was calculated by the modified scoring system described by Wang et al. and Hasan et al. (34, 35). The histology score was the sum of the following four parameters: size of alveolar spaces, thickness

of alveolar septa, alveolar fibrin deposition, and neutrophil infiltration (0: absent and appears normal; 1: light; 2: moderate; 3: strong; and 4: intense; total score is 16). The bacteria count in the lungs and BALF were analyzed by using LB plates. WBCs and PLTs in mice BALF were detected by the blood autoanalyzer and flow cytometry, cell precipitates were visualized by Swiss Giemsa staining, while cytokines in BALF and serum were measured by ELISA.

## Morphological analysis

To analyze the ultrastructure of *K. pneumoniae,* the samples were fixed with 2.5% and 3% glutaraldehyde respectively at 4°C for at least 12 h for slicing. The samples were sent to the center of electron microscopy in Fourth Military Medical University. Sections were examined with transmission electron microscopy (JEM-1230; Jeol USA, Peabody, MA, USA) and scanning electron microscopy (Hitachi S-3400N, Japan), and digital images were captured with a CCD camera (Olympus, Tokyo, Japan).

## Biofilm formation

The platelet-*K. pneumoniae* coculture supernatant incubated with *K. pneumoniae* in 24-well plates, and sterile cell slides were placed in each well. The cocultured liquid was incubated at 37°C for 7 days. 1640 RPMI medium was gently discarded, and an equal volume of fresh 1640 RPMI medium was slowly added along the side wall of the well every 24 h. Finally, the medium was discarded and washed once with sterile PBS; then, 1 mL of 3% glutaraldehyde was added and fixed samples at 4°C for 12 h. The all samples were sent to the center of electron microscopy in Fourth Military Medical University.

## Flow cytometry analysis

### Analysis of platelet activation by flow cytometry

Briefly, platelet was cultured with *K. pneumoniae* as described above. Then, (1 µL each) anti-Human CD62P (BD Biosciences, USA) were added into 19 µL platelets for staining and incubated in darkness at 22°C for 15 min. After incubation, washed the platelets with buffer and suspended in 300 µL of buffer. Finally, the sample was analyzed by flow cytometry.

### Analysis of phosphatidylserine exposure about K. pneumoniae

*K. pneumoniae* was cultured and washed as described above. The samples were diluted to 1:100 with $1 \times$ binding buffer. Then, Annexin V and PI (1 µL each) (FITC Annexin V Apoptosis Detection Kit I, BD Biosciences, USA) were added into 100 µL diluted cells for staining and incubated in darkness at 22°C for 15 min. After incubation, washed the cells with $1 \times$ binding buffer and suspended in 500 µL of $1 \times$ binding buffer. Finally, the sample was analyzed by flow cytometry.

### Detection of membrane depolarization

$DiBAC_4(3)$ (Sigma-Aldrich, St. Louis, MO, USA) was used to detect the membrane potential of *K. pneumoniae*. *K. pneumoniae* were cultured and washed as described above. $DiBAC_4(3)$ (0.1 µg/µL, 1 µL) was added to 1 mL diluted cells for staining. The cells were then incubated in darkness at room temperature for 30 min. After staining, the FITC-$DiBAC_4(3)$ fluorescence intensity of the samples was analyzed by flow cytometry.

Analysis of bacterial proteins with caspase-like substrate affinity. FITC-conjugated pan-caspase inhibitor peptide, Z-V-d-FMK (Intracellular Caspase Detection ApoStat, USA), was used to detect whether platelet-*K. pneumoniae* coculture supernatant could induce the expression of bacterial proteins that can bind to caspase substrate peptides. The cells of *K. pneumoniae* were cultured and washed as described above. Then, 10 µL Z-V-d-FMK-FITC was added into 1 mL diluted cells and incubated at 37°C in the dark for

30 min. Next, the cells were washed with PBS and suspended with 500 µL PBS. Finally, the fluorescence intensity of Z-V-d-FMK-FITC was analyzed by flow cytometry.

### Detection of K. pneumoniae DNA fragmentation

*K. pneumoniae* were added to platelet-*K. pneumoniae* coculture supernatant for 8 h. The samples were collected and DNA fragmentation was detected by TUNEL assay (*In Situ* Cell Death Detection Kit, Fluorescein, Roche, Mannheim, Germany). All steps were performed according to the manufacturer's instructions and analyzed by flow cytometry.

### Analysis of reactive oxygen species (ROS) by flow cytometry

*K. pneumoniae* were added to platelet-*K. pneumoniae* coculture supernatant for 8 h. The samples were collected, and the ROS level was detected by DCFH-DA (5 µM, KeyGen Biotech, Nanjing, China). All steps were performed according to the manufacturer's instructions and analyzed by flow cytometry.

### RNA analysis

After 8 h of coculture with or without platelet-*K. pneumoniae* coculture supernatant, RNA was extracted from *K. pneumoniae* using the RNeasy RNA isolation kit (Qiagen, Dusseldorf, Germany). The expression of five genes related to oxidative phosphorylation, viz. *ndhO, ndhR, qoxC, qoxD, azr*, was analyzed using qRT-PCR; 16S rRNA was used as the housekeeping gene. The sequences of primers used for analyzing other genes were as follows:

16S *rRNA,* F (5′-3′): ACGTGGATAACCTACCTATAAGACTGGGAT
R (5′-3′): TAACCTTACCAACTAGCTAATGCAGCG
*ndhO*, F (5′-3′): TGCGACCATGTTTCTTTGCG
R (5′-3′): GCCACGCTTACCATTTGACC
*ndhR*, F (5′-3′): CGTCGTTAATGCAGGCTGATG
R (5′-3′): CACGAAACGCATTGACTGGA
*qoxC*, F (5′-3′): CATCCGCTATACACCATCCCT
R (5′-3′):TCGCTAGGTATCGTTTGGGC
*qoxD*, F (5′-3′): TAAGTGTGAAGAGTGACCGCC
R (5′-3′): TTGGCTTTGCATTCGTCCAAG
*azr*, F (5′-3′): TTCAGTTTTATGCGGTTCGGC
R (5′-3′): TGCTGAAGGACCACAAGGTTT

### Statistical analysis

Statistical analyses were performed using GraphPad software (Prism v.5.01 La Jolla, CA, USA), and the results of flow cytometry were analyzed using FlowJoV7.6 software. All data are presented as mean ± standard error of the mean. The differences between the mean variables of the two groups were determined using Student's *t* test. One-way analysis of variance (ANOVA) with Tukey's post hoc test was used for multiple group analyses. Statistical significance was set at $P < 0.05$.

### ACKNOWLEDGMENTS

The authors thank the Clinical Laboratory of Xijing Hospital, Fourth Military Medical University of China, for technical assistance.

This research was supported by the National Natural Science Foundation of China (no. 82170226 and 81873448).

Wenting Wang and Yaozhen Chen conducted most of the experiments; Wenting Wang, Yaozhen Chen, Yutong Chen, and Erxiong Liu wrote the manuscript; Jing Li, Ning An, Jinmei Xu, Shunli Gu, Xuan Dang, and Jing Yi helped to perform experiments; Wen Yin, Xingbin Hu and Qunxing An designed and supervised the study.

## AUTHOR AFFILIATIONS

[1]Department of Transfusion Medicine, Xijing Hospital, Fourth Military Medical University, Xi'an, Shaanxi, China

[2]Faculty of Life Science College, Southwest Forestry University, Kunming, Yunnan, China

## AUTHOR ORCIDs

Wenting Wang ⓘ http://orcid.org/0000-0002-1638-4809
Qunxing An ⓘ http://orcid.org/0009-0008-9609-3307
Xingbin Hu ⓘ http://orcid.org/0000-0002-3952-3373
Wen Yin ⓘ http://orcid.org/0000-0002-7467-2286

## FUNDING

| Funder | Grant(s) | Author(s) |
| --- | --- | --- |
| MOST \| National Natural Science Foundation of China (NSFC) | 82170226, 81873448 | Wen Yin |

## ADDITIONAL FILES

The following material is available online.

### Supplemental Material

**Supplemental figures (Spectrum01279-23-s0001.pdf).** Fig. S1 to S10.

### Open Peer Review

**PEER REVIEW HISTORY (review-history.pdf).** An accounting of the reviewer comments and feedback.

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
