## [Reviewer comments · Microbiology Spectrum]

Microbiology Spectrum

Supernatant of Platelet-Klebsiella pneumoniae Coculture Induces Apoptosis-like Death in Klebsiella pneumoniae

Wenting Wang, Yaozhen Chen, Yutong Chen, Erxiong Liu, Jing Li, Ning An, Jinmei Xu, Shunli Gu, Xuan Dang, Jing Yi, Qunxing An, Xingbin Hu, and Wen Yin

Corresponding Author(s): Wen Yin, Fourth Military Medical University

Review Timeline:

Submission Date:	May 26, 2023
Editorial Decision:	September 2, 2023
Revision Received:	September 30, 2023
Accepted:	December 13, 2023

Editor: Fei Chen

Reviewer(s): The reviewers have opted to remain anonymous.

Transaction Report:

DOI: <https://doi.org/10.1128/spectrum.01279-23>

September 2, 2023

Prof. Wen Yin
Fourth Military Medical University
Department of Transfusion Medicine, Xijing Hospital
15#,Changle West Road,Xincheng district,Xi'an,Shaanxi province,China
Xi'an, Shaanxi 710032
China

Re: Spectrum01279-23 (Supernatant of Platelet-Klebsiella pneumoniae Coculture Induces Apoptosis-like Death in Klebsiella pneumoniae)

Dear Prof. Wen Yin:

Link Not Available

Sincerely,

Fei Chen

Journals Department
Reviewer comments:

Reviewer #1 (Comments for the Author):

1. In Figure 1A, the turbidity of the "SPLT+KP" group was significantly higher than that of the "KP" group. However, this difference is not reflected in Figure 1B, and the result is even reversed. When combined with the analysis of bacterial count results, turbidity detection does not appear to be a reliable method for determining inhibited bacterial growth.
2. The description of the experimental method for bacterial counting is unclear. The numbers of viable cells could be determined through serial dilution and plating assay.
3. The authors should assess the direct inhibitory effect of platelets on bacteria, rather than solely focusing on the platelet-K.

pneumoniae coculture supernatant.

4. The authors demonstrated that activated platelets could inhibit the growth of *K. pneumoniae*. However, the specific factor by which activated platelets impede bacterial growth remains unidentified. Only by uncovering the precise mechanism of action will the authors be able to make a better judgment regarding the emergence of a new antimicrobial strategy.

Reviewer #2 (Comments for the Author):

Comments to the Author

This study titled "Supernatant of Platelet-Klebsiella pneumoniae Coculture Induces Apoptosis-like Death in Klebsiella pneumoniae" revealed that the platelet-*K. pneumoniae* coculture supernatant can inhibit *K. pneumoniae* growth by inducing an apoptosis-like death. The study involves a significant amount of workload. The results are really interesting and meaningful, which has inspired the development of future antimicrobial strategies, and are expected to improve the clinical treatment of Gram-negative bacteria and control the development of multidrug-resistant strains.

My comments are as follows:

- (1) The turbidity of the four groups in FIG 1A is a little difficult to distinguish;
- (2) What are the corresponding time points for the results in FIG 1B and 1D? The bacterial counts at the three time points of 4h, 6h, and 8h among the three groups (Sn+KP/SPLT+KP/KP) are indeed contradictory between FIG 1C and FIG 1E (FIG 1C: similar; FIG 1E: Sn+KP, remarkably lower than the other two groups);
- (3) The mice were euthanized 48h post-injection. It will be advisable to consider extending the treatment duration further to investigate whether pneumonia symptoms of mice will completely alleviate (just a suggestion);
- (4) Line 298-299: The results showed that the ST group of CRKP was negative. What does 'negative' mean?
- (5) FIG S9: KPC should be revised as "*bla*KPC"; All gene names should be italicized.

Staff Comments:

Preparing Revision Guidelines

Please return the manuscript within 60 days; if you cannot complete the modification within this time period, please contact me. If you do not wish to modify the manuscript and prefer to submit it to another journal, please notify me of your decision immediately so that the manuscript may be formally withdrawn from consideration by Microbiology Spectrum.

Comments to the Author

This study titled “**Supernatant of Platelet-*Klebsiella pneumoniae* Coculture Induces Apoptosis-like Death in *Klebsiella pneumoniae***” revealed that the platelet-*K. pneumoniae* coculture supernatant can inhibit *K. pneumoniae* growth by inducing an apoptosis-like death. The study involves a significant amount of workload. The results are really interesting and meaningful, which has inspired the development of future antimicrobial strategies, and are expected to improve the clinical treatment of Gram-negative bacteria and control the development of multidrug-resistant strains.

My comments are as follows:

- (1) The turbidity of the four groups in FIG 1A is a little difficult to distinguish;
- (2) What are the corresponding time points for the results in FIG 1B and 1D? The bacterial counts at the three time points of 4h, 6h, and 8h among the three groups (Sn+KP/S_{PLT}+KP/KP) are indeed contradictory between FIG 1C and FIG 1E (FIG 1C: similar; FIG 1E: Sn+KP, remarkably lower than the other two groups);
- (3) The mice were euthanized 48h post-injection. It will be advisable to consider extending the treatment duration further to investigate whether pneumonia symptoms of mice will completely alleviate (just a suggestion);
- (4) Line 298-299: The results showed that the ST group of CRKP was negative. What does 'negative' mean?
- (5) FIG S9: KPC should be revised as “*bla*_{KPC}”; All gene names should be italicized.

Reviewer comments:

Reviewer #1 (Comments for the Author):

1. In Figure 1A, the turbidity of the "S_{PLT}+KP" group was significantly higher than that of the "KP" group. However, this difference is not reflected in Figure 1B, and the result is even reversed. When combined with the analysis of bacterial count results, turbidity detection does not appear to be a reliable method for determining inhibited bacterial growth.

Author Response: Thank the reviewer for pointing out this issue! We have examined Figure 1 carefully and repeated the bacteriostatic experiment. According to the results, we have re-adjusted Figure 1 as follows (Modified Fig.1). In our research, we set up four groups, including Sn+KP as the experimental group (Sn: the supernatant prepared after co-culture of bacteria and platelets), S_{PLT}+KP as the comparison group (S_{PLT}: the supernatant obtained by physical lysis of platelets), KP group without any treatment as a negative control, IPM group was treated with antibiotic as a positive control. Firstly, we detected the OD_{600nm} value of each group at 24 h and drew the bacterial growth curve (FIG. 1A). We found that the turbidity of Sn+KP group was significantly lower than that of S_{PLT}+KP group and KP group. Therefore, we detected and statistically analyzed the turbidity of each group of bacteria at 8 h (FIG. 1C and D). At the same time, we also found the same changes in bacterial counting after dilution of each group of bacteria at 2 h, 4 h, 6 h, 8 h, and 10 h (FIG. 1B). In the previous data, the photos shown in FIG. 1A were indeed not clear enough. We provided the original photo as follows and repeated the experiment again, as shown in FIG. 1C below.

We believe that the comparison of bacterial turbidity can directly reflect the proliferation of bacteria. In our research, we found that the turbidity of Sn+KP group was significantly lower than that of S_{PLT}+KP group and KP group, which was consistent with the bacterial counting. These results indicated that the antibacterial effect of the supernatant of activated platelet was significantly better than that of the supernatant of physical lysis of non-activated platelets. Detailed explanations are provided on Page 5 and 6, line 133 to 156.

Modified Fig.1

FIG 1. Platelet-*K. pneumoniae* coculture supernatant inhibits *K. pneumoniae* growth in vitro.

K. pneumoniae was added to the platelet-*K. pneumoniae* coculture supernatant. Sn+KP: *K. pneumoniae* cocultured with platelet-*K. pneumoniae* coculture supernatant. S_{PLT}+KP: *K. pneumoniae* cocultured with supernatant of platelet lysate, as a control group. KP: untreated *K. pneumoniae*, as a negative control. IPM: *K. pneumoniae* treated with Imipenem, as a positive control. (A) The growth curve of *K. pneumoniae* cocultured with or without platelet-*K. pneumoniae* coculture supernatant for 24 h, according to OD_{600nm}. (B) The counts of *K. pneumoniae* in each group at 2 h, 4 h, 6 h, 8 h, 10 h. (C) The photos of *K. pneumoniae* in each group at 8 h. (D) Analysis of OD_{600nm} in each group at 8 h. (E) The counts of *K. pneumoniae* in each group at 8 h. (F) Statistical results of the counts in (E). All results have been tested at least three times. Statistical analysis of the data was performed by one-way ANOVA, Tukey multiple comparisons.

original photo

2. The description of the experimental method for bacterial counting is unclear. The numbers of viable cells could be determined through serial dilution and plating assay.

Author Response: We supplemented the description of the experimental method for bacterial counting. The modifications are shown on Page 15, line 441 and 445.

3. The authors should assess the direct inhibitory effect of platelets on bacteria, rather than solely focusing on the platelet-*K. pneumoniae* coculture supernatant.

Author Response: We evaluated the direct inhibitory effect of platelets against *K. pneumoniae* in our previous research and found no significant change in bacterial solution turbidity after platelets acted directly on *K. pneumoniae* (as shown in “Supplementary Figure” below). In recent years, studies have shown that the antibacterial effect of platelets was mainly due to the release of antimicrobial peptides from platelets. The supernatants of platelets incubated with Gram-positive bacteria, such as *S. aureus*, have been reported to exert the same inhibitory effect on bacteria as platelets. We hypothesized that there might be differences in the direct antibacterial effect of platelets against Gram-positive and Gram-negative bacteria. In this study, to determine the inhibitory effect of platelets on Gram-negative bacteria *K. pneumoniae*, we extracted the platelet-*K. pneumoniae* coculture supernatant and then cocultured with *K. pneumoniae* to evaluate the inhibitory effect (as shown in FIG.1C below). The results showed that the platelet-*K. pneumoniae* coculture supernatant had obvious

antibacterial effect. Therefore, our research focused on the antibacterial effect of the supernatant of platelet-*K. pneumoniae* coculture.

Supplementary Figure

FIG 1C

PLT+KP: *K. pneumoniae* cocultured with platelet. **Sn+KP:** *K. pneumoniae* cocultured with platelet-*K. pneumoniae* coculture supernatant. **S_{PLT}+KP:** *K. pneumoniae* cocultured with supernatant of platelet lysate, as a control group. **KP:** untreated *K. pneumoniae*, as a negative control. **IPM:** *K. pneumoniae* treated with Imipenem, as a positive control.

4. The authors demonstrated that activated platelets could inhibit the growth of *K. pneumoniae*. However, the specific factor by which activated platelets impede bacterial growth remains unidentified. Only by uncovering the precise mechanism of action will the authors be able to make a better judgment regarding the emergence of a new antimicrobial strategy.

Author Response: We thank the reviewer for this kind suggestion. Our study provided evidence that the supernatant of platelet-*K. pneumoniae* coculture mediates the inhibition of *K. pneumoniae* growth. The inhibition mechanism was mainly explored from morphological changes, proliferation, apoptosis, oxidative stress damage, and metabolic damage after the interaction of bacteria and platelet-*K. pneumoniae* coculture supernatant. However, this study still has several limitations. For example, the detailed mechanism underlying the platelet-*K. pneumoniae* coculture supernatant-induced apoptosis-like changes in *K. pneumoniae* remains unclear. The special antibacterial components released by platelets after bacterial activation is a direction we would like to pursue in the future.

Reviewer #2 (Comments for the Author):

Comments to the Author

This study titled "Supernatant of Platelet-*Klebsiella pneumoniae* Coculture Induces Apoptosis-like Death in *Klebsiella pneumoniae*" revealed that the platelet-*K. pneumoniae* coculture supernatant can inhibit *K. pneumoniae* growth by inducing an apoptosis-like death. The study involves a significant amount of workload. The results are really interesting and meaningful, which has inspired the development of future antimicrobial strategies, and are expected to improve the clinical treatment of Gram-negative bacteria and control the development of multidrug-resistant strains.

The comments are as follows:

Reviewer #2

1. The turbidity of the four groups in FIG 1A is a little difficult to distinguish;

Author Response: We thank the reviewer for pointing out this issue. The photos shown in four groups are really not clear enough. The original photographs we provide are as follows. At the same time, we repeated the experiment and photographed again, as shown in Figure 1C.

original photo

FIG 1C

2. What are the corresponding time points for the results in FIG 1B and 1D? The bacterial counts at the three time points of 4h, 6h, and 8h among the three groups (Sn+KP/S_{PLT}+KP/KP) are indeed contradictory between FIG 1C and FIG 1E (FIG 1C: similar; FIG 1E: Sn+KP, remarkably lower than the other two groups);

Author Response: Thanks very much for pointing out our issue. We have examined FIG 1 carefully and repeated the bacteriostatic experiment. According to the results, we have re-adjusted FIG 1 as follows (Modified Fig.1). In our research, we set up four groups, including Sn+KP as the experimental group (Sn: the supernatant prepared after co-culture of bacteria and platelets), S_{P_{LT}}+KP as the comparison group (S_{P_{LT}}: the supernatant obtained by physical lysis of platelets), KP group without any treatment as a negative control, IPM group was treated with antibiotics as a positive control. Firstly, we detected the OD_{600nm} value of each group at 24 h and drew the bacterial growth curve (FIG. 1A). We found that the turbidity of Sn+KP group was significantly lower than that of S_{P_{LT}}+KP group and KP group. Therefore, we detected and statistically analyzed the turbidity of each group of bacteria at 8 h (FIG. 1C and D). At the same time, we also found the same changes in bacterial counting after dilution of each group of bacteria at 2 h, 4 h, 6 h, 8 h, and 10 h (FIG. 1B). Our original bacterial counting in Figure 1C shows the diluted pictures at 2 h, 4 h, 6 h, 8 h and 10 h, it may not reflect our results clearly, so we showed the plate count results by using the undiluted bacterial picture at 8 h (FIG. 1E) and the results of statistical analysis was shown in FIG. 1F. In our research, in order to prove the antibacterial effect of supernatant obtained by platelet activation was significantly better than that obtained by platelet physical lysis without bacterial activation, we compared the turbidity of Sn+KP and S_{P_{LT}}+KP groups. The explanation in detail was shown on Page 5 and 6, line 133 to 156.

Modified Fig.1

FIG 1. Platelet-*K. pneumoniae* coculture supernatant inhibits *K. pneumoniae* growth *in vitro*.

K. pneumoniae was added to the platelet-*K. pneumoniae* coculture supernatant. Sn+KP: *K. pneumoniae* cocultured with platelet-*K. pneumoniae* coculture supernatant. S_{PLT}+KP: *K. pneumoniae* cocultured with supernatant of platelet lysate, as a control group. KP: untreated *K. pneumoniae*, as a negative control. IPM: *K. pneumoniae* treated with Imipenem, as a positive control. (A) The growth curve of *K. pneumoniae* cocultured with or without platelet-*K. pneumoniae* coculture supernatant for 24 h, according to OD_{600nm}. (B) The counts of *K. pneumoniae* in each group at 2 h, 4 h, 6 h, 8 h, 10 h. (C) The photos of *K. pneumoniae* in each group at 8 h. (D) Analysis of OD_{600nm} in each group at 8 h. (E) The counts of *K. pneumoniae* in each group at 8 h. (F) Statistical results of the counts in (E). All results have been tested at least three times. Statistical analysis of the data was performed by one-way ANOVA, Tukey multiple comparisons.

3. The mice were euthanized 48h post-injection. It will be advisable to consider extending the treatment duration further to investigate whether pneumonia symptoms of mice will completely alleviate (just a suggestion);

Author Response: We thank the reviewer for kind suggestion. A mouse model of severe infection was established in our research. We initially set up three bacterial concentrations of 10⁴, 10⁶, 10⁸ CFU/mL. When mice were infected with low bacterial concentration, the symptoms of infection were completely relieved on day 7 without

any treatment. However, when the concentration of bacteria was 10⁸ CFU/mL, the mice suffered severe lung injury within 24-48 h and died within 72 h without any treatment. Compared with the infected group, the lung infection and injury of the mice treated with supernatant for 48 h were significantly improved, and the mice survived. Therefore, we selected this time point to treat the mice and evaluate its therapeutic effect.

4. Line 298-299: The results showed that the ST group of CRKP was negative. What does 'negative' mean?

Author Response: We thank the reviewer for pointing out this issue. We performed a string test on this clinically isolated resistant strain and the result was negative. This experiment helped us to identify the isolated strain as classical drug-resistant *K. pneumoniae* rather than hypermucoviscous phenotype (HvKP). We have modified the minor mistake in our manuscript. The modification is shown on Page 10, line 301 and 302.

5. FIG S9: KPC should be revised as "blaKPC"; All gene names should be italicized.

Author Response: We have revised KPC as "*bla_{KPC}*" and italicized the gene names in FIG S9.

Re: Spectrum01279-23R1 (Supernatant of Platelet-Klebsiella pneumoniae Coculture Induces Apoptosis-like Death in Klebsiella pneumoniae)

Dear Prof. Wen Yin:

Your manuscript has been accepted, and I am forwarding it to the ASM production staff for publication. Your paper will first be checked to make sure all elements meet the technical requirements. ASM staff will contact you if anything needs to be revised before copyediting and production can begin. Otherwise, you will be notified when your proofs are ready to be viewed.

Sincerely,
Fei Chen
Editor
Microbiology Spectrum

Reviewer #2 (Comments for the Author):

All the raised questions have been well supplemented and answered.